# Utilizing Electrochemical Biosensors as an Innovative Platform for the Rapid and On-Site Detection of Animal Viruses

**DOI:** 10.3390/ani13193141

**Published:** 2023-10-08

**Authors:** Xun He, Shan Wang, Caoyuan Ma, Guang-Ri Xu, Jinyou Ma, Hongbing Xie, Wei Zhu, Hongyang Liu, Lei Wang, Yimin Wang

**Affiliations:** 1Henan Institute of Science and Technology, Xinxiang 453003, China; xunhe1@hotmail.com (X.H.); sirui610@126.com (S.W.); caoyuanma6@hotmail.com (C.M.); xugr@hist.edu.cn (G.-R.X.); marsjy@163.com (J.M.); xhb9607@126.com (H.X.); zhuwei5983@hotmail.com (W.Z.); 2Shuangliao Animal Disease Control Center, Siping 136400, China; liuhongyang.com.cn@163.com; 3Ministry of Education Key Laboratory for Animal Pathogens and Biosafety, Zhengzhou 450046, China

**Keywords:** electrochemical biosensor, animal virus, detection, diagnostic assay, nucleic acid, antigen, antibody, aptamer

## Abstract

**Simple Summary:**

The detection of animal viruses remains a formidable scientific challenge, while concurrently presenting a profoundly consequential practical concern of considerable magnitude, necessitating the development of rapid, sensitive, specific, on-site, cost-effective, and user-friendly diagnostic assays. In response to this demand, electrochemical biosensors have garnered significant attention from researchers. In our review, we comprehensively assess the recent research progress pertaining to electrochemical biosensors for animal viral detection, with a particular focus on the application of screen-printed electrodes.

**Abstract:**

Animal viruses are a significant threat to animal health and are easily spread across the globe with the rise of globalization. The limitations in diagnosing and treating animal virus infections have made the transmission of diseases and animal deaths unpredictable. Therefore, early diagnosis of animal virus infections is crucial to prevent the spread of diseases and reduce economic losses. To address the need for rapid diagnosis, electrochemical sensors have emerged as promising tools. Electrochemical methods present numerous benefits, including heightened sensitivity and selectivity, affordability, ease of use, portability, and rapid analysis, making them suitable for real-time virus detection. This paper focuses on the construction of electrochemical biosensors, as well as promising biosensor models, and expounds its advantages in virus detection, which is a promising research direction.

## 1. Introduction

Viruses are a unique class of infectious, obligate intracellular parasites whose genetic material is composed of either DNA or RNA [1]. The virus particle itself is composed of a nucleocapsid, which contains the genome with the ability to replicate and a protein shell [2]. In the case of enveloped viruses, the nucleocapsid is surrounded by a lipid membrane resembling that of the host cell, which is studded with spike structures. The virus is a parasitic entity that relies on the host cell machinery to synthesize its own viral components, allowing for the successful replication and spread of the virus [3]. Viral infections pose a significant menace to both public health and animal husbandry [4,5,6]. Viruses can be transmitted through various means such as water, land, air, body fluids, and excreta, among others [7], and viruses can spread rapidly and eventually lead to the elimination or death of infected animals, resulting in significant economic losses [8,9,10].

The African Swine Fever virus (ASFV) has occurred in 22 countries worldwide since 2016, with Asia being the most affected in terms of animal losses. In China, the outbreak of ASF was first confirmed on 3 August 2018 [11]. According to the Chinese government’s website, a total of 143 cases of ASF were reported by July 2019 and more than 1.2 million pigs were culled. The estimated direct economic impact of ASF in China amounts to CNY 1 trillion, which is without considering the upstream and downstream of the industrial chain. The outbreak of ASF in Vietnam in 2019 had a severe impact on the Vietnamese pig sector, with more than 20% of the country’s pigs being culled or killed [12].

The impact of the virus is not only in the breeding of pigs but also in the poultry industry. In 2003, the Netherlands experienced an epidemic of highly pathogenic avian influenza (HPAI) caused by the H7N7 virus subtype. Thirty million birds were culled in this outbreak, about one third of the total poultry population [13]. The HPAI outbreak in Turkey from 2005 to 2006 resulted in a EUR 28 million loss for broiler producers [14]. There are other animal viruses, such as blue-ear disease in pigs [15], foot-and-mouth disease [16], Marek’s disease [17], etc. Therefore, it is important to reduce economic losses by the rapid detection of viruses.

The rising apprehension within the livestock industry regarding the emergence and dissemination of numerous animal viruses has prompted the adoption of diverse control strategies. These measures are designed to curtail the virus’s propagation and mitigate the associated losses [18]. However, effective viral detection remains a key factor in managing these pathogens. Traditional diagnostic methods such as nucleic acid amplification-based techniques [19], antigen or antibody-based assays [20], and viral isolation [21] all exhibit a number of limitations, including extended testing times, specialized equipment requirements, and technical expertise [22]. Therefore, there is an urgent need for fast, user-friendly, and field-applicable virus detection modes.

The electrochemical biosensor is an attractive platform for quick virus detection. Electrochemical sensors have proven to be an inexpensive and sensitive method, and are used to detect analytes involved in healthcare, environmental monitoring, and food packaging by diagnosing the virus before it spreads and cutting off transmission routes [23]. Traditional virus detection methods have lagged behind and are a significant cause of outbreaks. Electrochemical biosensors have heightened sensitivity and selectivity, affordability, ease of use, portability, and rapid analysis, making them suitable for real-time virus detection and overcoming the limitations of traditional detection methods [24]. Due to the scarcity of electrochemical biosensors for the rapid detection of animal viruses, the purpose of this review is to use the characteristics of electrochemical biosensors and their contributions in other fields to cite these achievements for the rapid detection of animal viruses (Figure 1).

## 2. Electrochemical Biosensors

### 2.1. Components of Electrochemical Biosensors

Each biosensor has four parts: sample or analytes, a biorecognition element, a transducer, and a signal processing device. Biosensors are categorized based on their transducer into electrochemical, optical etc. An electrochemical biosensor is a kind of detection device that uses biological molecules including proteins, nucleic acids, etc., to specifically bind to target analytes [25] (Figure 2A). The biological material is used as the sensitive element of the electrochemical biosensor, the electrode is used as the conversion element, the potential or current is the characteristic signal, and the characteristic signal is reflected in the analytical test device on the electrochemical workstation [26]. In electrochemical experiments, a commonly employed configuration is the three-electrode system, comprising a working electrode (WE), a reference electrode (RE), and a counter electrode (CE) [27]. The WE is the place where the electrochemical reaction occurs and is the object of study, the RE is used as a reference to measure and control the system potential, and the CE is used to complete the closed circuit to achieve the electrochemical measurement. In order to better connect the biomaterial with the electrode and exert the best performance of the sensor, the help of some compounds, such as thiol compounds [28] and conductive polymers [29], is usually needed to make the electrode fully modified (Figure 2B). Biomaterials were modified onto a WE to form complexes. The electrode–biomaterial complex binds to the analyte, and usually these bindings are irreversible, such as the paired hybridization of nucleic acids and the specific binding of antigens and antibodies, combined with different detection methods of the electrochemical workstation, and the behavior of electrode modifications in the above process is recorded on the electrochemical workstation [30].

#### 2.1.1. Electrodes

Commonly used electrode materials are gold electrodes, glass carbon electrodes, graphene electrodes, screen-printed electrodes, and so on (Figure 2C). The gold electrode is widely favored due to its inert chemical properties and its ability to combine non-chemically with mercaptan on the surface of the gold electrode to form gold-sulfur (Au-S) bonds with self-assembled monolayers [31], which has been widely demonstrated in the field of nucleic acid hybridization. The glassy carbon electrode is one of the most widely used working electrodes. It has good conductivity, high hardness, a wide polarization range, and can be used directly as an inert electrode for anodic dissolution. Graphene, characterized by a monolayer of SP2-bonded carbon atoms arranged in a honeycomb lattice, exhibits remarkable attributes such as rapid electron transfers, remarkable thermal conductivity, and favorable biocompatibility. Its superior physical and chemical properties render graphene an ideal electronic material for advanced electrochemical sensing applications [32].

Screen-printed electrodes (SPEs) are an ideal component for sensor applications. Compared with the traditional three-electrode system, the sensor surface is modified with nanomaterials to ensure consistency and stability while greatly simplifying the experimental operation process. SPEs generally include a substrate for printed electrodes. The substrate is printed with an external insulation layer and electrode leads, and the substrate is also printed with a WE, a RE, and a CE. Each electrode is connected with the corresponding lead to form a three-electrode system [33]. Because the low-cost manufacturing technology of SPEs can be easily extended to mass production, and all types of materials can be added to screen-printing slurry, SPEs can be customized for different substrate materials, shapes, and sizes in production to meet the needs of a variety of research. Its low manufacturing cost and ease of manufacture make it possible to mass customize personalized products according to demand, making it an ideal tool in the field of quality-controlled, scientific research [34].

#### 2.1.2. Working Stations

An electrochemical workstation is the abbreviation of the electrochemical detection system. Electrochemical workstations can be directly used to measure the steady state current on ultra-microelectrodes. After special treatment, if the workstation is connected with the micro-current amplifier and shielding box, a current of 1 pA or lower can be measured, which provides a solid foundation for electrochemical detection. Two primary categories of electrochemical workstations exist: single-channel workstations and multi-channel workstations. The difference is that multi-channel workstations can be used to detect multiple samples at the same time. At present, the electrochemical workstation is a commercial product, and different manufacturers provide different models of products with different electrochemical measurement techniques and functions. In order to meet the needs of rapid detection and be convenient to carry, a portable electrochemical workstation has been developed, which has the advantages of being small and light, easy to operate and cheap, and provides researchers with greater flexibility and convenience [35] (Figure 2D). Electrochemical biosensors can detect very small sample sizes and speed up the analysis process, making them powerful and highly sensitive devices [36]. So far, other types of sensors have also been reported, such as optical sensors [37], etc. Compared with other types of sensors, electrochemical biosensors do not require expensive equipment, save time, are convenient to carry, and are more user-friendly, especially for areas with backward economic development and scarce resources [24]. The common detection methods of electrochemical biosensors include voltammetry, amperometry, and the impedance method (Figure 2E).

### 2.2. Signal Analysis and Output

#### 2.2.1. Voltammetry

Voltammetry is based on the voltage change between the electrode and the electrolyte solution [38], of which cyclic voltammetry is the most commonly used electrochemical method and is considered “spectroscopy for electrochemists”. After triangular wave scanning, the electrode completes a reduction and oxidation process. In the range of scanning potential, the electrode can undergo alternate reduction and oxidation reactions, and the resulting current–voltage curve is recorded, reflecting the steady-state response of chemical reactions triggered by electron transfers [39]. Cyclic voltammetry can be used in a faster time, has a wider potential to see the electrode where the Oxidation reduction reaction occurs, and provides a rich signal for the electrode. Cyclic voltammetry is generally used for qualitative analyses and rarely in quantitative analyses.

#### 2.2.2. The Ampere Method

Amperometric detection relies on the alteration of currents. A fixed potential is applied to the electrochemical solution, which is enough to oxidize or reduce a certain electroactive substance in the solution. The change of the current and time is recorded to obtain the current–time curve [40,41]. The ampere method is an electrochemical analysis method to study the kinetics of electrode processes.

#### 2.2.3. The Impedance Method

The electrochemical impedance spectra (EIS) theory and its data interpretation are very complex for researchers who are not familiar with it, such as biologists, biochemists, or materials scientists. The impedance method is an electrochemical measurement technique that employs a low-amplitude sine wave potential (or current) as a perturbation signal [42]. The reaction rate depends on the frequency, and the frequency change can distinguish the reaction rate of different substances in the solution. Due to its broad measurement frequency range, the impedance method provides access to a greater wealth of dynamic information and electrode interface structural details compared to other conventional electrochemical methods [43].

These basic electrochemical methods all play a crucial role in the signal changes of the working electrode modification process. On the basis of these methods, some more sensitive methods have evolved, such as pulse voltammetry, step voltammetry, the multi-point step chronoelectric method, etc., and a foundation has been established for more novel electrochemical biosensor models with potential applications.

## 3. The Electrochemical Biosensor Model

The earliest prototype of electrochemical biosensors can be traced back to the 1970s, when the sensor was based on the specific capture of the analyte by affinity elements and the subsequent conversion of the capture event into a measurable signal, aiming to broaden the selectivity and sensitivity of analyte detection in the field of electrochemical detection [44]. Due to the advantages of low cost and being fast and convenient, electrochemical biosensors have been widely considered and are more suitable for commercial applications [45]. The first glucose sensor was reported in 1962, and a glucose sensor using Clark technology was commercialized in 1975 [46]. In the developments since then, electrochemical glucose biosensors have generated great commercial value [47]. Electrochemical biosensors have been widely studied and applied to human society to promote the development of society, including environmental detection [48,49], food safety detection [50,51], human pathogenic microorganisms detection [52,53], etc., but there is little research on animal viruses.

The impact of animal viruses on animal husbandry is huge, and the economic losses caused by animal viruses are inestimable. People are growing increasingly concerned and are attempting to apply the established advancements in human virus detection to the study of electrochemical biosensors. The goal is to swiftly detect animal viruses and minimize the economic losses they cause. Electrochemical biosensors can be used to detect animal viruses mainly based on nucleic acids and proteins. Available models for the detection of animal viruses by electrochemical biosensors are described below.

### 3.1. The Nucleic Acid-Based Electrochemical Biosensor Model

Electrochemical biosensor platforms are based on biological molecules that recognize each other through various interactions and cause an electron transfer to produce electrical signals to complete detection. Nucleic acids are not only the basic genetic material of organisms but are also the ideal material for the identification of various analytes [54]. The specificity of the pairing of nucleotide molecules that constitute nucleic acids further shows the advantages of nucleic acid-based electrochemical biosensors.

The reduction of DNA by electrochemical sensors was reported as early as 1960 [55], and subsequently, since DNA purine bases can be oxidized by electrochemistry, electrochemical sensing methods have been developed for the indirect oxidation of DNA by metal complexes acting as electrochemical mediators [56]. With the development of electrochemical biosensors, nucleic acid has gradually become a bridge between electrode materials and analytes, which also benefits from the easy modification of DNA. Therefore, a promising electrochemical biosensor has been developed, in which the Oxidation reduction reaction of electroactive substances upon the surface of the electrode is caused by specific hybridization events of ssDNA, and the electrochemical signal is generated by transferring electrons. This method is popular in the detection of human pathogens, food guarding, the environment, etc., and can also be used to detect animal viruses.

This type of sensor immobilizes ssDNA to the working electrode by both chemical and non-chemical binding, depending on the specificity between the modified nucleic acid and the electrode material. I. Macwan et al. created a DNA electrochemical biosensor with biotin modification to detect biological macromolecules. They deposited the avidin monomer onto GO and investigated the interaction between the avidin monomer and GO. Biotin-modified DNA forms stable complexes through the hydrophobic force between avidin and graphene, including van der Waals forces, hydrogen bonds, and electrostatic interactions [57]. The biotin-modified ssDNA hybridized with the complementary ssDNA in the target analyte, and the hybridization signal was converted to an electrical signal to complete the detection of DNA-binding proteins (Figure 3A). This is the first report on the interaction between graphene and avidin. Avidin has the ability to specifically bind to biotin, which is easily labeled by nucleic acids. This study shows a new method for connecting graphene with nucleic acid molecules and provides a new idea for the detection of animal viral nucleic acids or nucleic acid-labeled proteins. More prevalent is the thiolation modification of DNA, which is attached to the gold working electrode by Au-S [58], There have been many studies on the detection of human viruses [59,60]. The ssDNA fixed on the surface of the electrode hybridizes with the complementary DNA, interacts with RNAs and proteins, or produces corresponding electrical signals through structural changes of the DNA (Figure 3B). The electrical signals can also be generated indirectly by the recognition process of ssDNA using enzymes and Oxidation reduction reaction mediators. The reaction was combined with the electrochemical method on an electrochemical workstation. There are many similar electrochemical biosensors based on nucleic acids. Zhao et al., using gold nanomaterials as electrodes, detected CTDNA through an Au-S connected sandwich model, with a linear range of 1 × 10^−15^ to 1 × 10^−8^ mol/L and a detection limit as low as 5 × 10^−16^ mol/L [61]. Lin et al. modified gold nanoparticles on iron-rich nanotubes and then immobilized the probe with a gold-S bond and completed the detection of Mir-486-5P in A549 cells, and the detection limit could be as low as 8.53 × 10^−16^ mol/L [62]. The characteristics of these electrochemical DNA biosensors include being portable, simple, cost-effective, having a fast response time, high sensitivity, and high selectivity, and their compatibility with miniature detection technologies has garnered significant interest in electrochemical DNA biosensors. Such biosensors are deemed feasible for the detection of animal viruses.

### 3.2. The Antibody- and Antigen-Based Electrochemical Biosensors Model

Proteins constitute a vital component of viruses and play a crucial role in their life cycle [63]. Proteins have a protective effect on viruses but also determine the specificity of a virus infection, for the host, and the virus protein is also an important antigen. There is potential in using viral particles and viral surface proteins to make electrochemical biosensors for the detection of viruses. Several electrochemical biosensors have been reported as exemplary models for detecting SARS-CoV-2 nucleocapsid and spike proteins [64,65,66]. In recent years, electrochemical biosensors based on the mutual recognition of antibodies and antigens have attracted attention, which are related to immunoassay technology and have the basic characteristics of immunoassay. Antibody- and antigen-based electrochemical biosensors integrate the high sensitivity and specificity of immunoassay with the conversion of physical signals. When the antibody–antigen immune reaction occurs, this specific binding to form a complex is usually irreversible, and after the transduction of the electrochemical workstation signal, the recognition signal is transformed into an electrical signal, which makes it possible to detect viruses rapidly.

The key to antigen detection is the fixation of the antibody with the electrode, and usually the antibody is directly fixed to the electrode surface. Various methods enable the attachment of antibodies to the electrode surface. Adsorption allows for the direct attachment of antibodies to the electrode surface. M. Veerapandian et al. devised an electrochemical biosensor utilizing the methylene blue adsorption of GO (graphene oxide) to detect the antigen of the influenza A virus [67]. Chitosan was modified to GO electrodes containing methylene blue (MB) for the direct adsorption of protein A. Protein A is a specific receptor for fc antibodies and can directly immobilize monoclonal antibodies (Figure 4A). MB, as the electroactive, was adsorbed onto the GO electrode by cyclic voltammetry (20 cycles). Chitosan and protein A were modified to the surface of GO–MB by incubation at room temperature. Monoclonal antibodies were subsequently adsorbed as probes for influenza antigens. The sensor’s generated current, subsequent to the binding of the antibody and antigen, exhibits a robust correlation with the antigen’s concentration, with a detection limit as low as 10 pM for H5N1 and H1N1.

The immobilization of antibodies to electrodes can also be achieved through a matrix of conductive polymers (CPs), which are mainly used as sensors for biological interactions. The ability of high conductivity to monitor the electrical signal generated by the interaction of the probe with the analyte is key to its preference. Polypyrrole is the most commonly used conductive polymer due to its biocompatibility [68], high hydrophilic properties [69], and high stability in water. Studies on the biotinylated single-chain variable fragment antibody and functionalized polypyrrole have shown that the combination of polypyrrole and biotin-streptavidin is an effective method for immobilizing antibodies [70]. Two functional monomers, poly (pyrrole) and 3-N-hydroxyphthaldimethylpyrrole, were electropolymerized to the gold electrode to complete the electrode modification. Biotin was covalently bound to the functional poly (pyrrole) membrane, and streptavidin was incubated, followed by the biotin binding antibody. Casein was used as a blocking agent to avoid non-specific binding (Figure 4B). This study showed that the standard curve of the sensor was linearly related to a 1 pg/mL–100 ng/mL antigen concentration, was highly reproducible, and the detection limit had the potential to reach an impressive 1 pg/mL.

There are other ways to immobilize antibodies, such as a special membrane structure called self-assembling monolayers (SAMs) [71]. Self-assembled monolayers (SAMs) refer to organic layers that spontaneously form upon the electrode’s surface when molecules form a solution or when gas phases are adsorbed. This membrane structure has a strong choice of terminal functional groups and has the potential to be applied in the construction of nanoscale electrochemical biosensors by immobilized antibodies [72]. The adsorption of alkane thiols on gold is the most widely used SAMs application. The 5′ thiol-modified and 3′ biotin-modified ssDNA is connected to the gold surface by self-assembly, and the affinity of biotin and streptavidin is used to form the Au-ssDNA-biotin-streptavidin electrochemical biosensor body. The biotin-labeled antibody is attached to the sensor body, which is also a potential method for immobilizing antibodies. From a protein chemistry point of view, immobilizing proteins on the electrode surface is an extremely complex problem [73,74]. Moreover, the protein is sensitive and has low conformational stability, which may be destroyed upon immobilization on the electrode surface. By using SAMs and oligonucleotides to immobilize antibodies, a bond between antibodies and electrodes can be established, which can effectively avoid the problems of antibody ring breaking, inactivation, and non-specific adsorption.

From another point of view, the body will also produce a resistant response to the invasion of the virus, and the corresponding antibodies will be produced in the blood, so the detection of virus antibodies can also provide a basis for the possible existence of the virus. At this time, the antigen of the virus needs to be fixed, and the possible virus can be detected by combining the antigen with the antibody in the blood to generate an electrical signal [75]. This type of electrochemical biosensor is similar to the antibody-based electroactive biosensor, which will not be described here.

### 3.3. The Nucleic Acid Aptamer-Based Electrochemical Biosensor Model

Nucleic acid aptamer is a kind of artificial biological receptor, which is composed of a small sequence of oligonucleotide nucleotides or short peptides obtained by in vitro screening. An artificial bioreceptor was independently discovered by Ellington and Szostak [76] and Tuerk and Gold [77] in 1990, and a peptide aptamer was further introduced by Colas et al. [78] in 1996. The discovery of a nucleic acid aptamer as a new bioreceptor provided a new research direction for electrochemical biosensors based on nucleic acid aptamers. As a new biological receptor, aptamers have unique advantages. First nucleic acid aptamers were screened and synthesized in vitro by SELEX (systematic evolution of ligands by exponential enrichment) [79,80] technology, and further large-scale production using chemical synthesis methods can reduce costs while also overcoming the use of animals or cells where antibody production is necessary. Due to the short oligonucleotide nucleic acid, a nucleic acid aptamer is easier to be modified by chemical groups, is easier to connect with the electrode, and has greater openness. At the same time, it also reduces the difficulty of antibodies connecting to the electrode and simplifies the experimental process. Secondly, aptamers have a stronger ability to bind to targets than antibodies, which is due to the fact that aptamers can use multiple action sites to bind to targets through hydrogen bonding, electrostatic interactions, van der Waals forces, etc. [81]. For some targets that cannot be effectively bound by antibodies, aptamers can effectively solve this problem. Nucleic acid aptamers have the ability to specifically bind to the target of antibodies and are better than antibodies in some properties, so they are suitable for electrochemical biosensing.

Similar to other types of electrochemical biosensors, nucleic acid aptamer-based electrochemical biosensors also generate signals by transferring electrons from electroactive substances, which has been widely studied for targeted drug deliveries and biomarker detection. The same principle can also be applied to the detection of animal viruses. An ingeniously designed sensor model takes advantage of the structural changes before and after the aptamer binds to the target and uses the electroactive probe attached to the aptamer at a distance from the electrode for electron transfers. Aptamer conformation changes lead to a change in the spatial separation between the electrically active probe and the electrode, which in turn produces a change in the signal [82,83]. This is a hairpin-like structure design, and the conformational change of the nucleic acid aptamer can meet the rapid detection requirements. Firstly, one end of the ssDNA is modified with a sulfhydryl group, and the other end is modified with an electroactive substance. The sulfhydryl end can self-assemble to the surface of the gold electrode to complete the ligation of the DNA and electrode. When there is no substance to be tested, the electroactive substance is far away from the surface of the gold electrode, and no electron gain or loss is generated, that is, no electrochemical signals. When there is a substance to be tested, because the substance to be tested is specific to the designed ssDNA, the specific binding will occur at this time, causing a structural change of the ssDNA. As a result, the electroactive substance will approach the surface of the gold electrode, generating electron gains and losses, that is, generating electrochemical signals (Figure 5A). Similar to this, Yang et al. fixed an ssDNA and gold electrode through an Au-S bond and used AuNPs to interact with the ssDNA non-covalently. After binding, they served as a conductive bridge. At this time, the ssDNA structure changed, and an electron transfer occurred between the electroactive substance and the electrode in the solution, resulting in electrochemical signals, and the detection of DNA associated with the breast cancer gene (BRCA1) was as low as 1 pM [84]. In another common design, a nucleic acid aptamer acts as a link between the electrode and the target analyte. Binding of the aptamer to the target analyte prevents the electroactive substance from moving closer to the electrode to transfer electrons, which in turn increases the impedance [85,86]. This impedance design has faster reaction kinetics, can ignore interferences due to the background current, and has higher sensitivity. It is preferred to design the ssDNA with a hairpin structure that is self-assembled to the surface of the gold electrode by sulfhydryl groups, and the void sites between the ssDNA are closed with other short-chain thiols to avoid non-specific bindings. When there is no substance to be measured, the electrochemical substance in the solution can contact the electrode surface, and the electron gain and loss occur to produce an electrochemical signal. When there is a substance to be tested, the surface of the electrode is covered due to the specific binding of the ssDNA to the substance to be tested. At this time, the electroactive substance in the solution is hindered and will be reduced or will be unable to contact the electrode surface. As a result, the electrical signal is reduced or disappeared, and the corresponding impedance will increase (Figure 5B). Similarly, Yeter et al. used a glass fiber-carbonized electrode to design an impedance sensor for detecting HIV-1 DNA. They used a glass fiber-carbonized electrode that was chemically modified to connect gold nanoparticles through the electrode and then connected the thiol-modified ssDNA to gold nanoparticles. At this time, the electroactive substances in the solution could freely contact the surface of the electrode. When the ssDNA and target DNA were complementary, the electroactive substance in the solution could not contact the electrode surface due to a steric hindrance. At this time, the impedance increased, and the detection limit could be as low as 13 fM (1.3 × 10^−14^ M) [87].

## 4. Advantages and Limitations of Electrochemical Biosensors for Virus Detection

### 4.1. Advantages

The advantages of an electrochemical biosensor in virus detection are its short detection time and convenience, the fact that it is cheap and simple, and its real-time detection [88]. Usually, the virus is diagnosed in the laboratory by virus isolation, PCR, ELISA, and other methods. The accuracy and practicability of these traditional methods are beyond doubt, which have left a strong mark in the history of the human fight against viruses. However, these methods require specialized personnel, expensive equipment, and testing lags [89]. Due to these limitations, the detection of viruses by electrochemical biosensors has come to its final stage in history. Table 1 shows the comparison of virus detection by conventional methods and electrochemical biosensors.

On the other hand, the popularity of electrochemical biosensors also benefits from the maturity of SPEs. SPEs have long been considered the most promising analytical tool in electrochemical detection. Compared with traditional electrodes, an SPE avoids the polishing, cleaning, and activation required for other solid electrodes and greatly simplifies the experimental process. Commercial SPEs have a high versatility, which is also one of the advantages of SPEs [96,97]. Because the low-cost manufacturing technology of SPEs can be easily extended to mass production and all types of materials can be added to screen-printing slurry, SPEs can be customized for different substrate materials, shapes, and sizes in production to meet the needs of a variety of research. At the same time, SPEs are small, powerful, low-cost, and maintenance-free and have been widely used in electrochemical research in environmental monitoring [98,99,100], clinical diagnoses [101,102], drug analyses [103], and food detection [97,104,105], which has promoted the development of electrochemical biosensors. The wide application of nanomaterials is also the cause of the rapid development of electrochemical sensors. We all know the characteristics of nanomaterials, including their small size effect, surface and interface effect, quantum size effect, etc. Gold and carbon nanomaterials are commonly used in the field of electrochemistry. These nanomaterials can be used as nanostructured electrodes for the amplification of electrical signals, thereby improving detection sensitivity [106,107]. They can also be used to prepare signal labels such as AuNPs [108] and can also be used as a platform for highly conductive nanostructures such as graphene, graphene oxide, carbon nanotubes, gold nanometers, etc., to obtain an extremely low detection limit [109].

### 4.2. Limitations

The advantages of electrochemical biosensors, such as their being fast, convenient, and cheap, have indeed made them more and more prominent in the diagnostic industry, although some electrochemical biosensors are still in the development and testing stages. The current problems faced by electrochemical biosensors are mainly the following aspects. The first is accuracy and reproducibility, and analyte concentration becomes increasingly important when moving target analyte testing from a clean laboratory with a controlled environment to the field. Here, many external environmental factors can contribute to significant differences in signal strength, such as the temperature, humidity, the sample volume, the electrode surface area, non-calibrated instruments, and contamination. In response to this problem, corresponding coping strategies have been reported to improve the accuracy and reproducibility of biosensors through ratio electrochemistry [110]. Secondly, the limited shelf life and stability of the biometric components as well as non-specific binding are still the biggest biosensor limitations, and corresponding strategies have been reported to overcome and reduce these aspects. Finally, the concentration used in the sensor component design is trace and requires precise operations; therefore, it is only through a reasonable design and rigorous testing that biosensors can be transferred from the laboratory to the field.

## 5. Conclusions

Rapid detection of pathogens, such as viruses, remains a great challenge in analytical medicine due to their quantitative diversity [111]. The economic loss caused by viruses is inestimable every year. The development of a rapid, cost-effective, user-friendly, and highly specific virus detection strategy is of utmost urgency. Therefore, cost-effective, fast, convenient, and sensitive electrochemical biosensors have been favored [112]. Due to the rapid detection characteristics of electrochemical biosensors and their characteristics of biological material constructions, which will pave the way for the identification of different viruses, the next stage of development should focus on the universality of electrochemical sensors and the detection of viruses with high accuracy, followed by their ability to adapt to work in multiple environments. With the development of materials science, the popularization and application of new materials, and the rapid development of electrochemical components, the rapid detection of viruses by electrochemical biosensors will be a valuable depression in the future. In summary, the development of electrochemical biosensors mainly relies on being fast, convenient, specific, and cost-effective. Considering these properties, combining biosensors and biomanufacturing approaches with synthetic biology approaches, or combining all these principles, will be key to a successful development of robust biosensors in the future.

## Figures and Tables

**Figure 1 animals-13-03141-f001:**
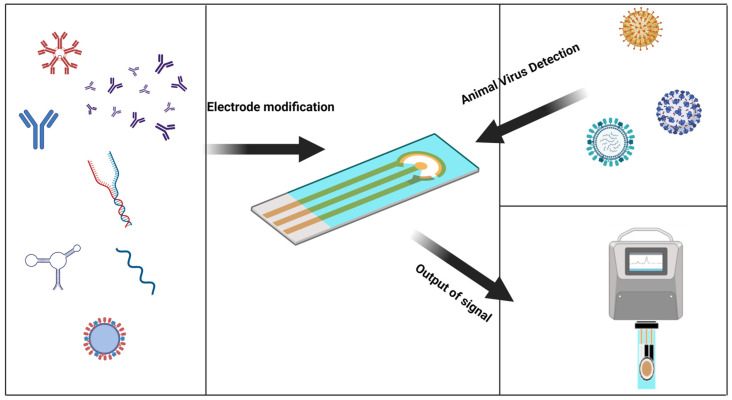
Graphical abstract, a general introduction to the electrochemical sensor detection process. (Created with BioRender.com, accessed on 17 September 2023).

**Figure 2 animals-13-03141-f002:**
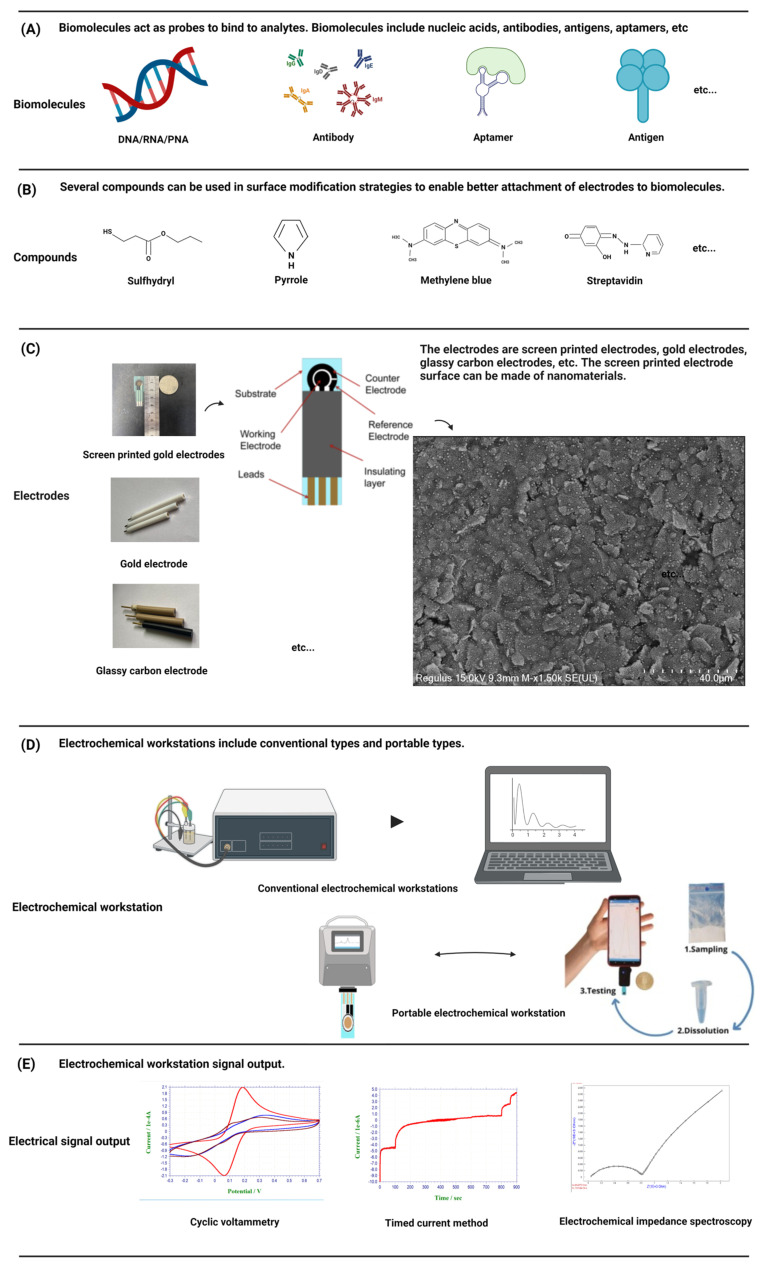
A schematic diagram of a standard electrochemical workstation and portable electrochemical workstation is presented. The components of the detection system, including electrodes, biomolecules, compounds, etc., and the conventional signal output are introduced. (Created with BioRender.com, accessed on 17 September 2023).

**Figure 3 animals-13-03141-f003:**
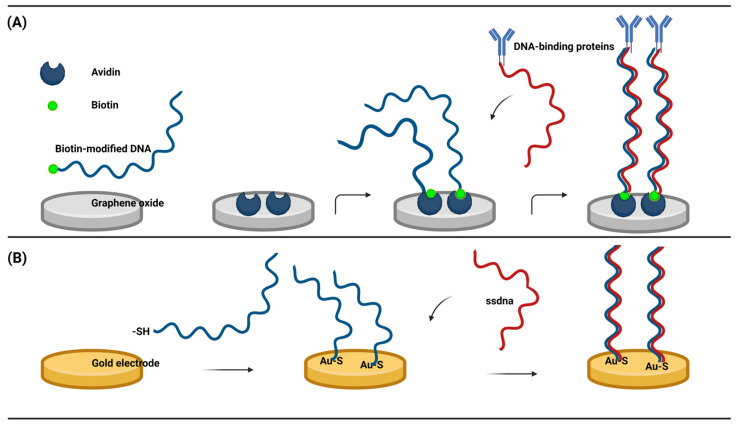
(**A**) Graphene and avidin formed polymers by a hydrophobic force to link biotin-modified ssDNAs and detect DNA assembly proteins. (**B**) Gold electrodes were ligated with sulfhydryl modified DNA to detect complementary ssDNAs. (Created with BioRender.com, accessed on 17 September 2023).

**Figure 4 animals-13-03141-f004:**
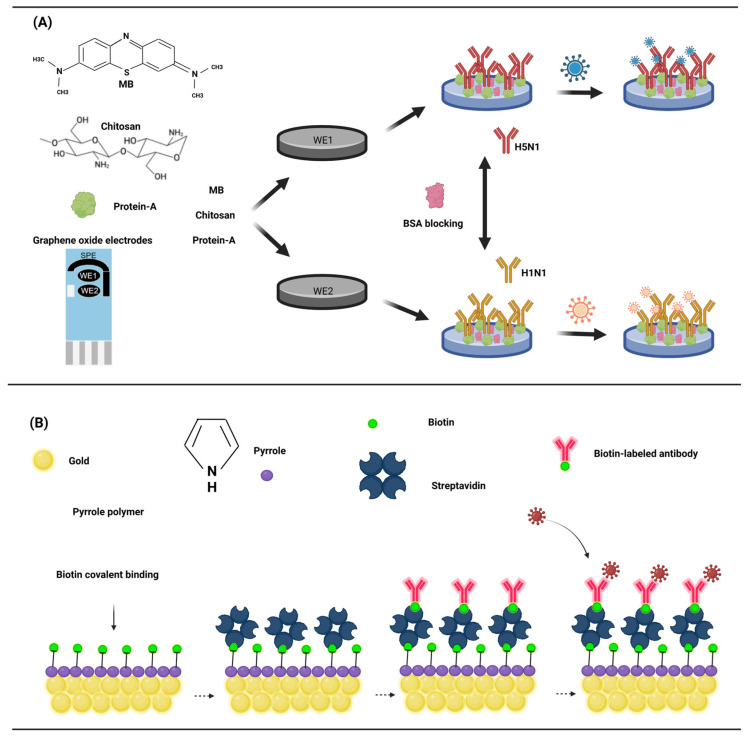
(**A**) The GO surface is rich in a large number of oxygen-containing functional groups that adsorb methylene blue, and protein A was adsorbed by chitosan. The influenza virus was detected by a protein A-linked influenza virus antibody. (**B**) After the electro polymerization of pyrrole on the surface of the gold electrode, biotin was covalently bound, and streptavidin was used as a bond to adsorb the biotin-labeled antibody. (Created with BioRender.com, accessed on 17 September 2023).

**Figure 5 animals-13-03141-f005:**
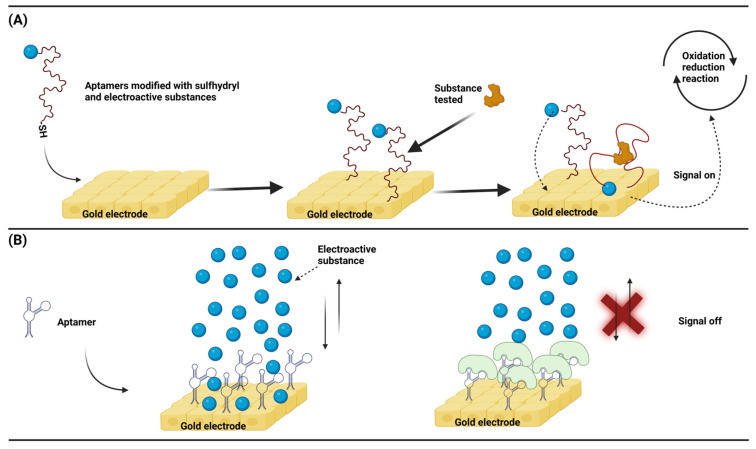
(**A**) The spatial structure of the aptamer changes after binding to the analyte, and the analyte detection generates an electrical signal by the proximity of the electroactive substance to the electrode. (**B**) Upon binding of the analyte to the aptamer, the electroactive species free in the solution cannot approach the electrode and the impedance increases. (Created with BioRender.com, accessed on 17 September 2023).

**Table 1 animals-13-03141-t001:** Comparison of virus detection methods.

Methods	Time Required	Convenience	On-Site Detection	Linear Range	LOD	References
Virus isolation	2–3 Days	Cannot be carried	No	/	/	
PCR	1–2 h	Cannot be carried	No	/	2.52 × 10^1^ copies/µL	[90]
				/	10 CFU/ML	[91]
ELISA	6–8 h	Cannot be carried	No	/	3.675 × 10^4^ copies/µL	[92]
				0.5 × 10^−15^–5.0 × 10 ^−6^ g/ML	0.5 × 10^−15^ g/ML	[93]
Electrochemical biosensor	10–36 min	Is portable	Yes	1.176 to 4.825 μg/mL	3.569 × 10^1^ ng/mL	[94]
				1–1 × 10^3^ pfu/mL	1 pfu/mL	[95]

## Data Availability

This article is a literature review; therefore, all data described here were obtained from the research cited in the references.

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
