# Peer review of "Utilizing Electrochemical Biosensors as an Innovative Platform for the Rapid and On-Site Detection of Animal Viruses"

_animals, 2023, doi:10.3390/ani13193141_

Round 1
Reviewer 1 Report
In this review, the research team discusses the research progress of electrochemical biosensing in the diagnosis of animal viral diseases, and the direction of future efforts. This review will provide reference for the development of novel diagnostic techniques for animal viral diseases. However, the article still needs to be further modified and perfected. Please revise it carefully according to the opinion.
1. n line 22, “a promising tool” should replaced by “promising tools”.
2. Line 39 and line 41 should in one paragraph.
3. Line41-43, this sentence is too complicated, it is better to revise and make it easy to understand.
4. Line 5, African Swine Fever virus (ASF) should be “African Swine Fever virus (ASFV) ”.
5. The author should use the abbreviation form of African Swine Fever in line 47-49.
6. Line 60-61, this sentence is difficult to understand.
7. Line 75, should delete “a”.
8. Line 80, 100, 126, 154, should capitalize the first letter.
9. Line 95, “, ”should replaced by “.”.
10. Line 113, should deleted “are”.
11. Figure 1, the figure legend should be in detail, should illustrated Figure 1A-E separately.
12. Figure 1A, the word is not clearly.
Author Response
We have revised the reviewer's comments, please see the attachment.

Reviewer 2 Report
1) It would be better to combine the simple summary and abstract if the journal’s style allows it.
2) The aim or purpose of the review should be described at the end of the introduction.
3) I would strongly suggest to the authors to add at line 80: each biosensor has four parts: sample or analytes, biorecognition element, transducer, and signal processing device. Biosensors are categorized based on their transducer into electrochemical, optical…
4) It would be nice if the authors provide a graphical abstract at the end of the introduction.
5) Electrochemical techniques (lines 154-187) have been described briefly. Everyone in the field of biosensors knows almost all of this information. I suggest deleting this part and present this information as a table to compare these electrochemical techniques and pay more attention to the new approaches to detect animal viruses (the main part of the review)
6) What is the difference between graphene oxide and gold in Figure 2? Which substrate has more accurate results?
7) The authors did not discuss the application of nanotechnology (nanoparticles, nanorods) to develop electrochemical biosensors to get lower LOD and higher specificity.
8) The authors did not provide the latest commercially available technologies to detect animal viruses. They could compare the latest articles in a table with focusing on LOD, specificity, and antibody/aptamer using along with their reference.
9) The authors should compare four methods described in Table 1 in LOD and specificity also with appropriate reference.
10) The authors could read and potentially cite this paper for brucella abortus detection using an electrochemical approach.
https://www.sciencedirect.com/science/article/pii/S266683192300036X
Author Response

(The authors gave the same response as above.)

Reviewer 3 Report
The present manuscript provides an overview of the utilization of electrochemical biosensors as an innovative platform for the rapid and on-site detection of animal viruses. I have identified specific areas of concern concerning the current manuscript:
1. Despite its focus on electrochemical biosensors for diagnosing animal viruses, the manuscript lacks any mention or discussion of published data regarding chemical biosensors for the same purpose. Is there a dearth of available literature on this topic?
2. Merely outlining the detection methodology and the materials employed in chemical biosensors, which are not elaborated upon in sufficient detail, falls short of the comprehensive approach expected in a review article.
3. Several instances within the manuscript exhibit incomplete paragraphs, and some content appears obscured beneath the figures. This suggests that the authors may be inclined to publish incomplete information rather than make a concerted effort to provide essential details to the scientific community.
4. The quality of the figures in the manuscript is suboptimal, and there is a need to break them down into multiple figures to align with the level of detail mentioned in the text. This restructuring would enhance the manuscript's clarity and readability.
Minor editing is required
Author Response

(The authors gave the same response as above.)

Round 2
Reviewer 2 Report
The authors provided my comments well and I would recommend publishing the MS.
Reviewer 3 Report
I do not have any more comments on the manuscript.
Minor editing require